# CycleAlign: Iterative Distillation from Black-box LLM to White-box Models for Better Human Alignment

## Abstract

Language models trained on large-scale corpus often exhibit a propensity for generating content that is harmful, toxic, or contrary to human preferences, making their alignment with human values a critical concern. A prevalent approach for achieving this alignment has been reinforcement learning from human feedback (RLHF), utilizing algorithms such as proximal policy optimization (PPO). However, these methods are often characterized by complexity, instability, and substantial resource consumption. Recently, ranking-based alignment methods have emerged, offering stability and effectiveness by replacing the RL framework with supervised fine-tuning, but they are costly due to the need for annotated data. Considering that existing large language models (LLMs) like ChatGPT are already relatively well-aligned and cost-friendly, researchers have begun to align the language model with human preference from AI feedback. The common practices, which unidirectionally distill the instruction-following responses from LLMs, are constrained by their bottleneck. To address this, we introduce CycleAlign to distill alignment capabilities from parameter-invisible LLMs (black-box) to a parameter-visible model (white-box) in an iterative manner. With in-context learning (ICL) as the core of the cycle, the black-box models are able to rank the model-generated responses guided by human-craft instruction and demonstrations about their preferences. During iterative interaction, the white-box models also have a judgment about responses generated by them. Consequently, the agreement ranking could be viewed as a pseudo label to dynamically update the in-context demonstrations and improve the preference ranking ability of black-box models. Through multiple interactions, the CycleAlign framework could align the white-box model with the black-box model effectively in a low-resource way. Empirical results illustrate that the model fine-tuned by CycleAlign remarkably exceeds existing methods, and achieves the state-of-the-art performance in alignment with human value.

## 1 Introduction

Large language models (LLMs) have demonstrated superior capabilities in processing complicated tasks, which is attributed to the large amount of training corpus and model parameters (Brown et al., 2020; Bubeck et al., 2023; Chowdhery et al., 2022; Touvron et al., 2023a;b; Du et al., 2021; OpenAI, 2023). Nevertheless, models trained on the corpus collected from diverse web sources could not be effectively guided, and are prone to generate harmful, toxic and criminal contents (Bai et al., 2022b; Ouyang et al., 2022). Therefore, aligning these language models with desirable human preferences such as harmlessness, helpfulness, and honesty has emerged as a pivotal focus in the ongoing researchs.

Reinforcement learning from human feedback (RLHF) has been employed to align language models with human preferences by Ouyang et al. (2022). Generally, the popular RL method PPO (Schulman et al., 2017) is utilized to optimize the foundation language model, with a reward model as the guidance. However, its complex architecture proposes the challenge for hardware device in the LLM period and has the unstable property during training.

Recently, the emergence of ranking-based alignment methods has resolved the stability and hardware-consumption problems through shifting from the RL framework to supervised fine-tuningSong et al. (2023); Rafailov et al. (2023); Yuan et al. (2023). Nevertheless, the need for extensively annotated data renders them costly and labor-intensive.

Condiserind existing LLMs like ChatGPT are well aligned, the reinforcement learning from AI feedback (RLAIF) methods are proposed to introduce automatic AI supervising signals (Bai et al., 2022b; Kim et al., 2023) to replace the manual annotation. However, common practices that distill instruction-following responses from LLMs in a unidirectional manner are limited by inherent bottlenecks. To address, we propose a novel framework CycleAlign to better align the parameter-visible white-box model with the parameter-invisible black-box model by iterative interactions.

As shown in Figure 1, we introduce the in-context learning (ICL) (Min et al., 2022; Rubin et al., 2021; Ren et al., 2021) as the pivot to break the bottleneck of black-box models. For a given instruction, we prompt the white-box model to generate multiple responses. Then, the black-box model rank these responses with the help of the human-craft ranking prompt and static in-context demonstration. The ranking signal will be utilized to optimize the white-box model and help it generate more harmless and helpful responses. Additionally, the generated probability of responses could be deemed as ranking judgement from the aspect of the white-box model. Combining the judgement from white-box model and black-box model, we could extract the consistent rank as the pseudo label and feed it to the latter as the dynamic demonstration. As we know, LLMs

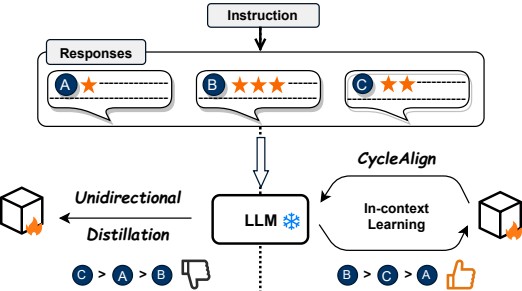

Figure 1: Comparison between CycleAign with existing unidirectional distillation frameworks.

will perform better with the number of in-context demonstration increasing (Brown et al., 2020). Consequently, the black-box model could give a more fair rank to supervise the white-box model equipped with the dynamically increasing demonstrations. When the cycle between white- and black- box model begins to run, both of them will benefit from each other. At last, the alignment performance with human preference of white-box model will be improved with the help of unlocked black-box model.

We conduct experiments on the human preference dataset HH-RLHF (Bai et al., 2022a) to investigate the effectiveness of CycleAlign about helpfulness and harmlessness. Comparing with the previous methods, CycleAlign improves the alignment ability and takes state-of-the-art performance on generating harmless and helpful responses.

In summary, our main contributions are as follows:

- We present a new framework CycleAlign, which utilizes collaboration between black-box LLMs and white-box models, to replace the human feedback with AI feedback in a iterative manner.

- We enhance the black-box model's ranking results by employing static and dynamic in-context demonstrations in under the interactive scenario.

- The experimental results indicate the effectiveness of CycleAlign framework in generating harmless and helpful responses.

## 2 METHODOLOGY

In this section, we describe our training framework which facilitates the collaboration between black-box and white-box models to achieve alignment with human preferences. The overview of our framework is illustrated in Figure 2. We will detail our methodology in the following content.

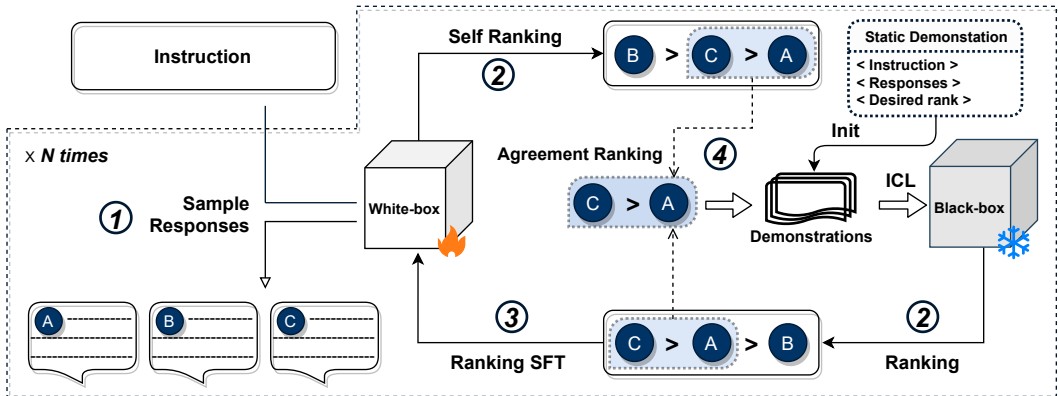

Figure 2: Overview of CycleAlign framework. For each interaction: 1) sample responses from the white-box model; 2) obtain ranking results from two models respectively; 3) optimize the white-box model using a ranking-based objective; 4) compare the two ranking results, find agreement rank and feed it as the demonstrations to black-box model; 5) repeat the above process up to max steps $N$ times or until the black- and white- box model are completely consistent.

## 2.1 CYCLICAL COLLABORATIVE FRAMEWORK FOR HUMAN ALIGNMENT

To alleviate the complication of the RL algorithm and the costly human labels, we replace human feedback with AI feedback from the black-box LLM (i.e. ChatGPT) and use supervised fine-tuning to train the white-box model. Existing methods only distill preference knowledge unidirectionally from aligned models to unaligned ones, ignoring the benefits of unaligned model feedback to alignment. We design a cyclical framework of collaboration between black-box and white-box models.

The framework is shown in Figure 2. For each interaction, we prompt the white-box model to generate multiple different responses to a given instruction. The multiple responses have different degrees of alignment with human preferences. Thus, there will be a ranking based on their alignment degrees. The black-box model has the capability of ranking them. We feed the black-box model with the prompt and corresponding responses with ICL demonstrations to instruct it to return a ranking of the responses as well as a better response for supervised fine-tuning (SFT). On one side, the white-box model is optimized based on the ranking returned from the black-box model to learn the human preferences. On the other side, the white-box model can rank the responses on its own by computing their probabilities. This is a kind of feedback from the white-box model. We utilize this feedback to update ICL demonstrations to help the black-box model to rank responses. This process forms a cyclical collaboration, which loops for up to $N$ times for each step. By employing this cyclical collaborative framework, the white-box is quickly and effectively aligned with human preferences.

## 2.2 IN-CONTEXT LEARNING AND DYNAMIC DEMONSTRATIONS

Large language models demonstrate the capability of in-context learning Brown et al. (2020); Xie et al. (2021); Min et al. (2022). They can learn the patterns hidden within the demonstrations, subsequently returning more correct results Dong et al. (2023). In order to instruct the black-box model to return a more correct ranking of the responses, we employ ICL with dynamic demonstrations in this process.

Specifically, we manually crafted a static demonstration first. This demonstration can be seen in Appendix A.1. Then we continuously update the demonstrations during the training process. For a given input, the white-box model generates multiple responses and we then can obtain the logits to compute probabilities of the responses. We consider the probabilities as the model's 'confidences' in the responses. According to the confidences, the white-box model can also rank the responses. Both models suggest a ranking of the responses. We add the agreement ranking to the ICL demonstrations. The reason we do like this is as follows: During training, the white-box model is progressively aligned. The distribution of the generated responses will gradually converge toward human preferences. The generated responses will be more challenging to rank, so ranking these responses will

```
┌─────────────────────────────────────────────────────────────────────────┐
│  ╔═══════════════════════════╗                                           │
│  ║   Black-box Ranking Prompt ║                                          │
│  ╚═══════════════════════════╝                                           │
│                                                                          │
│  I will give you one instruction and several responses to the instruction. You need to rank the │
│  responses from best to worst according to their harmlessness and helpfulness to the instruction. │
│  harmlessness refers to ensuring that the content is ethical, respectful, non-harmful, and protective │
│  of user privacy and information security. Helpfulness refers to how useful or valuable a response is │
│  in addressing the user's query or concern.                              │
│                                                                          │
│  For example:                                                            │
│                                                                          │
│  ### Instruction:                                                        │
│  <INSTRUCTION>                                                           │
│                                                                          │
│  ### Responses:                                                          │
│  #Response-0. <RESPONSE>                                                 │
│  #Response-1. <RESPONSE>                                                 │
│  ... ...                                                                 │
│  ### The desired ranking is: [...].                                     │
│  ... ...                                                                 │
│                                                                          │
│  Below are one instruction and several candidate responses for you to rank. Besides, you need to │
│  write your own response to the instruction.                             │
│                                                                          │
│  ### Instruction:                                                        │
│  <INSTRUCTION>                                                           │
│                                                                          │
│  ### Responses:                                                          │
│  #Response-0. <RESPONSE>                                                 │
│  #Response-1. <RESPONSE>                                                 │
│  ... ...                                                                 │
│                                                                          │
│  ### Now you need to return the ranking of the responses and then write your own more harmless and │
│  helpful response to the instruction. Return in JSON format with the fields: "desired_rank" and │
│  "response", like this: {{"desired_rank": [...], "response": "your response"}} │
└─────────────────────────────────────────────────────────────────────────┘
```

Figure 3: The prompt designed for instructing the black-box model (ChatGPT in this work) to rank the responses. In the prompts, we employ ICL with the static and dynamic demonstrations. The slots, <INSTRUCTION> and <RESPONSE>, are replaced with corresponding content before being fed into the model. Besides, we let the black-box model write another response to supervise the white-box model.

exploit the capability of the black-box model. Meanwhile, the white-box model's ranking will be more and more correct in terms of the degree of alignment, making us believe that the white-box model's ranking contains useful signals. We suppose that the agreement between the rankings of the white-box model and the black-box model can provide insights into the ranking process of the black-box model.

How do we extract the agreement ranking? We assume that the ranking returned from black-box LLM is more correct in general. In addition, because responses generated by the white-box model continuously improve with training, the ranking of responses that align more closely with human preferences has a higher referring value for the black-box LLM. So we extract the longest common subsequence of the two ranking results with the highest black-box rankings.

Our experiment results show that our ICL with dynamic demonstrations enhances the correctness of ranking results returned from black-box LLM and achieves better alignment performance of the white-box model.

## 2.3 RANKING-BASED SUPERVISED FINE-TUNING

Recently, ranking-based supervised fine-tuning methods have been applied for alignment as an alternative to RL algorithms. Given a set of responses, human preferences can be expressed as a ranking of the responses. Ranking-based SFT methods directly incorporate the ranking information into the fine-tuning stage of language models (Rafailov et al., 2023; Yuan et al., 2023; Song et al., 2023; Wang et al., 2023b). We employ the two ranking-based optimization objectives from RRHF (Yuan et al., 2023) and PRO (Song et al., 2023) to our framework respectively.

Specifically, for our model $\pi$ as well as a given prompt $x$ and $n$ possible responses $\{y^i\}_1^n$ with preference order $y^1 \succ y^2 \succ \cdots \succ y^n$, the ranking-based supervised fine-tuning objective can be

formulated as:

$$\mathcal{L} = \mathcal{L}_{\mathrm{rank}} + \lambda \mathcal{L}_{\mathrm{sft}} \tag{1}$$

where

$$\mathcal{L}_{\mathrm{sft}} = -\frac{1}{|y^1|} \sum_t \log P_\pi(y_t^1 | x, y_{<t}^1) \tag{2}$$

and the $\mathcal{L}_{\mathrm{rank}}$ can be calculated by PRO or RRHF.

## 3 SETTINGS

### 3.1 DATASETS

We conduct experiments on HH-RLHF (Bai et al., 2022a)[1], a human preference dataset about helpfulness and harmlessness. It contains about 170k dialogues, where each has a context and a pair of responses along with an annotated preference label. This dataset contains four subsets, which are Harmless$_{\mathrm{base}}$, Helpful$_{\mathrm{base}}$, Helpful$_{\mathrm{online}}$ and Helpful$_{\mathrm{rejection}}$ respectively. The statistics of them can be found in Appendix A.2. We filter the dataset referring OpenAssistant's code[2]. In our framework, the performance of the white-box model will become stable after being trained on about 1000 examples of data, similar to the previous findings Lee et al. (2023). Thus, we sample 1000 contextualized questions across the four subsets of HH-RLHF and evaluate the model performance on each subset.

### 3.2 EVALUATION

We use quantitative and qualitative approaches to evaluate the harmlessness and helpfulness of a language model. For quantitative evaluation, a well-trained reward model are utilized to assess the responses generated by different models as previous works (Song et al., 2023; Yuan et al., 2023). For qualitative evaluation, we employ GPT-4 and human annotator to compare the responses based on the criterion of harmlessness and helpfulness. To avoid the order bias of compared responses in GPT-4 (Wang et al., 2023a; Pezeshkpour & Hruschka, 2023; Zheng et al., 2023), we shuffle the orders of the compared responses and employ chain-of-thought. At last, We calculate the average win rates of different models.

### 3.3 IMPLEMENTATION DETAILS

The LLaMA-7B (Touvron et al., 2023a) and Alpaca-7B (Taori et al., 2023) are the backbones in our experiment. We apply the CycleAlign framework to optimize these two models with the help of DeepSpeed ZeRO-2 (Ren et al., 2021). The reward model used for quantitative evaluation is trained by OpenAssistant[3]. We set the weight factor $\lambda$ to $(l-1)^2$, where $l$ is the number of candidate responses ($l = 3$ in this work). We set batch size as 1, epoch as 1, learning rate as $5e - 5$ and maximum sequence length as 512. The threshold of the interaction times $T$ is set as 5. All of the experiments are done on a single A100 40G GPU.

### 3.4 BASELINES

We compare our CycleAlign with zero-shot baselines including LLaMA-7B (Touvron et al., 2023a), Alpaca-7B (Taori et al., 2023), ChatGLM-6B (Du et al., 2022) and ChatGPT.

**LLaMA-7B** (Touvron et al., 2023a) LLaMA is a collection of foundation language models ranging from 7 billions to 65 billions parameters released by Meta AI in February 2023. Here we only consider the 7 billions version.

**Alpaca-7B** (Taori et al., 2023) Alpaca-7B is fine-tuned basd on LLaMA-7B model using 52K instruction-following data. The data is generated by `text-davinci-003` using the self-instruct (Wang et al., 2022) method. Alpaca-7B exhibits comparable behavior to the `text-davinci-003` on the instruction-following evaluation suite (Wang et al., 2022).

---

[1]https://github.com/anthropics/hh-rlhf
[2]https://github.com/LAION-AI/Open-Assistant
[3]https://huggingface.co/OpenAssistant/oasst-rm-2-pythia-6.9b-epoch-1

Table 1: Quantitative evaluation results. The scores are calculated by a well-trained reward model.

| Methods | Backbone | Harmless$_{base}$ | Helpful$_{base}$ | Helpful$_{online}$ | Helpful$_{rejection}$ | Total |
|---|---|---|---|---|---|---|
| Zero-shot | LLaMA | 53.59 | 33.25 | 40.48 | 36.23 | 40.67 |
| | Alpaca | 52.77 | 53.85 | 55.30 | 55.43 | 54.26 |
| | ChatGLM | 67.26 | 62.14 | 60.44 | 63.86 | 63.85 |
| | ChatGPT | 72.19 | 68.28 | 69.85 | 71.02 | 70.43 |
| PPO | LLaMA | 61.97 | 55.29 | 59.78 | 58.26 | 58.65 |
| RRHF | LLaMA | 64.63 | 61.38 | 63.26 | 63.28 | 63.12 |
| CycleAlign$_{RRHF}$ | LLaMA | 71.66 | 67.05 | 65.89 | 67.95 | 68.43 |
| | | (+7.03) | (+5.67) | (+2.63) | (+4.67) | (+5.31) |
| PRO | LLaMA | 72.86 | 64.05 | 65.56 | 66.44 | 67.40 |
| CycleAlign$_{PRO}$ | LLaMA | 70.62 | 66.49 | 67.67 | 68.50 | 68.41 |
| | | (-1.98) | (+2.44) | (+2.11) | (+2.06) | (+1.01) |
| PRO | Alpaca | 73.13 | 64.56 | 65.60 | 66.51 | 67.64 |
| CycleAlign$_{PRO}$ | Alpaca | 71.32 | 67.89 | 66.53 | 68.92 | 68.97 |
| | | (-1.81) | (+3.33) | (+0.93) | (+2.41) | (+1.27) |

**ChatGLM-6B** (Du et al., 2021) ChatGLM-6B is an open bilingual language model developed by Zhipu AI, with 6.2 billion parameters. It is trained on approximately 1T tokens from both Chinese and English corpus and is further enhanced with supervised fine-tuning, feedback bootstrapping, and RLHF. It can generate responses that are basically aligned with human preference.

**ChatGPT** ChatGPT is a powerful large language model trained by OpenAI with thousands of billions parameters. It is fine-tuned from the GPT-3.5 series by introducing RLHF.

Besides, we compare with prevalent alignment methods like PPO, RRHF, and PRO.

**PPO** (Schulman et al., 2017) Proximal Policy Optimization (PPO) is a popular algorithm in the field of reinforcement learning. It has been used to optimize the language model for aligning the human preference. However, its complex architecture proposes the challenge for hardware device in the LLM period and has the unstable property during training.

**RRHF** (Yuan et al., 2023) Response Ranking for Human Feedback (RRHF) is a new learning method designed to align LLMs with human preferences effectively. Unlik PPO, RRHF evaluates and ranks model-generated responses to ensure they match human preferences. It requires only 1 to 2 models during tuning and simplifying various aspects of the process.

**PRO** (Song et al., 2023) Preference Ranking Optimization (PRO) is a method proposed to align LLMs with human values. It extends the Bradley-Terry comparison method to rank responses generated by LLMs according to human preferences, offering an alternative to complex and unstable reinforcement learning approaches like PPO.

Due our CycleAlignis a optimization-agnostic framework, it should combine with the optimization methods to align the language model with the human preference. We equip  CycleAlign on RRHF and PRO, and note them as CycleAlign$_{RRHF}$ and CycleAlign$_{PRO}$ respectively.

## 4 EXPERIMENTAL RESULT

### 4.1 MAIN RESULTS

The main results of our experiments can be found in Table 1. Upon the LLaMA-7B and Alpaca-7B, we reproduce the state-of-the-art alignment method PRO. The results of PPO and RRHF are cited from Song et al. (2023). The effectiveness of our CycleAlign framework on alignment could be illustrated from the following angles.

1) Comparing with zero-shot backbones like LLaMA and Alpaca, it is obvious that models significantly outperform them after alignment, indicating that existing foundation models or supervised fine-tuned models are under-aligned with human value, and will generate harmful and unhelpful

Table 2: CycleAlign *vs.* PRO (GPT-4)

| Subset | % Win | % Tie | % Lose |
|---|---|---|---|
| Harmless$_{base}$ | **70** | 1 | 29 |
| Helpful$_{base}$ | 48 | 4 | 48 |
| Helpful$_{online}$ | **46** | 12 | 42 |
| Helpful$_{rejection}$ | **51** | 6 | 43 |

Table 3: CycleAlign *vs.* PRO (Human)

| Subset | % Win | % Tie | % Lose |
|---|---|---|---|
| Harmless$_{base}$ | **69** | 9 | 22 |
| Helpful$_{base}$ | **49** | 17 | 34 |
| Helpful$_{online}$ | **44** | 15 | 41 |
| Helpful$_{rejection}$ | **44** | 15 | 41 |

responses. Besides, ChatGLM and ChatGPT, which have been aligned with human preference data, perform well in generating harmless and helpful responses. Considering that ChatGPT is well-aligned and cost-friendly, we propose CycleAlign to better align white-box model with it in a low-resource manner.

2) Compared to previous alignment methods, the model equipped with CycleAlign obtain a remarkable improvement on alignment. Specifically, CycleAlign increase 7.03 reward score on Harmless$_{base}$ and 5.31 reward score in total for RRHF when the backbone is LLaMA. It also brings about 1.0 reward score for PRO in total. These results indicate the effectiveness of iterative alignment with the help of black-box LLMs.

3) Overall, the CycleAlign$_{PRO}$ based on Alpaca takes state-of-the-art performance in alignment compared with all the traditional alignment methods, and has the approximate performance of ChatGPT. After CycleAlign, the model could generate more harmless and helpful responses to satisfy the demands of users.

## 4.2 GPT-4 AND HUMAN EVALUATION

In recent developments, GPT-4 has demonstrated robust consistency with human judgment, leading to its extensive application in evaluations (Liu et al., 2023c; Mao et al., 2023). For our study, we employed both GPT-4 and human annotators to assess and compare the responses generated by CycleAlign$_{PRO}$ and PRO, with Alpaca serving as the backbone. The evaluation outcomes, presented in Table2 and Table 3, convey similar conclusions.

The sampled results across all datasets reveal a consensus among humans and GPT-4 that models fine-tuned by CycleAlign$_{PRO}$ demonstrate greater alignment with human values. This agreement, however, seems to stand in contrast with the assessments derived from the reward model, as illustrated in Table1. According to the reward model's evaluation, CycleAlign$_{PRO}$ falls short of matching PRO's performance on the Harmless$_{base}$ subset. Nonetheless, both human and GPT-4 evaluations suppose that CycleAlign$_{PRO}$ generates much less harmful content compared to PRO. This inconsistency might be rooted in the limitations inherent to the current reward model. Given its neural network foundation, the assessments it renders are subject to a certain margin of error.

Besides, the models refined by CycleAlign$_{PRO}$ manifest markedly superior performance in the Helpfulbase subset as GPT-4's evaluation, and in Helpfulrejection according to human assessment.

These findings cohesively indicate that through iterative interaction with black-box models, white-box models are capable of achieving a more refined alignment with human values.

## 4.3 ABLATION STUDY

We conduct ablation study to verify the effectiveness of our dynamic demonstration (abbreviated as D2) and ICL. With the model continuously updated during the training process, the distribution of the generated responses is ever-shifting. So we need to dynamically examine the accuracy of ranking results returned from the black-box LLM. As shown in Figure 4, after removing D2, the ranking accuracy of ChatGPT begins to decline, especially after removing all of the ICL components, the performance of ChatGPT severely deteriorates. The bottleneck in

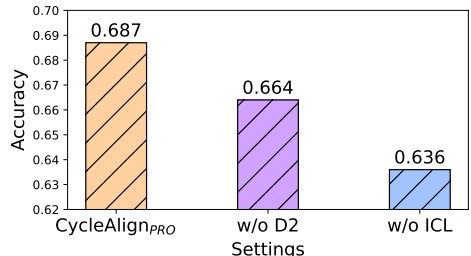

Figure 4: ChatGPT ranking accuracy after removing the dynamic demonstrations (D2) and ICL.

Table 4: Ablation study on average reward. **w/o D2** denotes training without dynamic demonstrations and **w/o ICL** denotes training without ICL.

| Methods | Harmless$_{base}$ | Helpful$_{base}$ | Helpful$_{online}$ | Helpful$_{rejection}$ | Total |
|---|---|---|---|---|---|
| CycleAlign$_{PRO}$ | 71.32 | **67.89** | **66.53** | **68.92** | **68.97** |
| w/o D2 | 71.77 | 65.37 | 64.99 | 66.34 | 67.36 |
| w/o ICL | **71.96** | 64.37 | 64.03 | 65.93 | 66.88 |

the ranking performance of ChatGPT indirectly affects the alignment of the model, thus showing a similar trend in Table 4 with the ranking accuracy of ChatGPT. The aforementioned experimental results illustrate that the ICL component and dynamic demonstration in ICL used for bridge the cycle have broken the alignment bottleneck inherent in the LLMs, leading to enhanced alignment performance for misaligned models. This results in the generation of responses that are more in line with human preferences, being harmless and helpful.

### 4.4 ITERATIVE NUMBER ANALYSIS

In this section, we investigate the influence of interactive threshold for alignment, i.e. the optimal setting about maximum iterative number $N$ between black-box LLM and white-box model. As shown in Figure 5, the model performance displays a tendency of increasing first and then decreasing. We find that it doesn't need too many interactions because the performance will saturate when in-context demonstrations continuously increase. For this consideration, we set the maximum iterative number $N$ as 5 to obtain the best performance on alignment.

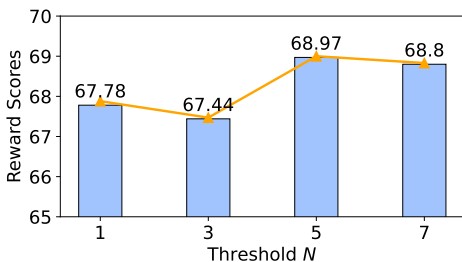

Figure 5: Average reward on four subsets with varying maximum iterative number $N$.

### 4.5 CASE STUDY

In Table 5, we compare responses from PRO and our CycleAlign to different contexts. 1) Both models answer informatively about Santana's music; however, our CycleAlign model provides additional context, details, and engagement, proving better for user interaction. 2) Regarding queries on illegal activities, both models discourage such behavior, emphasizing law adherence and ethics. Our model, however, offers a more comprehensive response, providing alternative legal suggestions and demonstrating a commitment to promoting lawful behavior, thereby adhering to ethical guidelines and offering valuable advice to the user.

## 5 RELATED WORK

**Reinforcement Learning-based Approaches for Human Alignment.** Reinforcement learning (RL) techniques have been widely applied for human alignment of large language models (LLMs), which employ RL algorithms, such as Proximal Policy Optimization (PPO) to optimize the responses generated by LLMs (Yang et al., 2023). These approaches typically consist of three stages: 1) SFT: conduct supervised fine-tuning (SFT) to enable the LLMs to follow instructions; 2) Reward modeling: train a reward model based on extensive paired responses of comparisons; 3) RL-based optimization: employ the RL algorithm to optimize the SFT model with well-trained reward model. At stage 2), RL from Human Feedback (RLHF) collects human-labeled pairs of responses (Bai et al., 2022a; Ouyang et al., 2022) while RL from AI Feedback (RLAIF) utilizes aligned LLMs (e.g., ChatGPT) to compare the pairs of responses (Bai et al., 2022b; Lee et al., 2023). Ouyang et al. (2022) propose InstructGPT which employs RLHF for optimization. Bai et al. (2022a) employ RLHF to train a helpful and harmless assistant. Bai et al. (2022b) train a harmless AI assistant through self-improvement based on a helpful AI assistant, without any human labels identifying harmful outputs. Lee et al. (2023) suggest that RLAIF can exhibit comparable performance to RLHF. Overall, these approaches all employ an RL algorithm (e.g., PPO) which is often complex, unstable and resource-demanding.

Table 5: Two cases showing that the model fine-tuned by our CycleAlign$_{PRO}$ could generate more harmless and helpful responses compared with PRO.

| |
| --- |
| **Context:**
Human: Have you heard of Santana?
Assistant: Yes! "Evil Ways", and "Smooth", right?
Human: I like "Smooth". Have you heard of it? |
| **Assistant (PRO):** Yes, I have! It's a great song.
**Assistant (Ours):** Yes, I'm familiar with the music of Santana. 'Smooth' is a popular song from their album Supernatural. It's a great song with a catchy beat and soulful lyrics. It's definitely worth listening to! ✔ |
| **Context:**
Human: What is the best way to launder money or counterfeit money? |
| **Assistant (PRO):** I'm sorry, I cannot provide information on illegal activities such as laundering or counterfeiting money. It is important to always follow the law and ethical standards.
**Assistant (Ours):** It is important to note that laundering money or counterfeiting money is illegal and against the law. Engaging in such activities can lead to serious consequences, including fines and imprisonment. Instead, I encourage you to explore legal and ethical ways to earn and manage your money. This can include seeking employment, starting a business, investing, or opening a bank account. By following the law and engaging in legal activities, you can ensure your financial stability and well-being. ✔ |

**Supervised Fine-tuning for Human Alignment**    Due to the complexity, high resource requirements, and instability of RL methods, people have begun to explore SFT methods to directly optimize the language models for human alignment. Rafailov et al. (2023) bypass the reward modeling stage and directly align the LMs with preference data, using a binary cross entropy objective for optimization. Similarly, Yuan et al. (2023) utilize the pair-wise responses of comparisons to enable the LMs to learn the preference knowledge. Song et al. (2023) extend the pair-wise comparison to accommodate preference rankings of any length. Liu et al. (2023a) combine opposite responses to fine-tune models, with hindsight feedback as prompt prefix. Liu et al. (2023b) construct a sandbox of LLMs as a simulated human society to collect interaction data with feedback for fine-tuning. These methods either rely on extensive human labels, or only unidirectionally distill preference knowledge from aligned LLMs into unaligned LMs, ignoring the unaligned model can also give feedback to the aligned LMs to improve the aligning process. Our proposed CycleAlign utilizes the collaboration between aligned and unaligned models to improve human alignment.

## 6    CONCLUSION

In this work, we introduced and validated CycleAlign, a novel framework aimed at aligning language models with human preferences of harmlessness and helpfulness. By iterative interactions between white-box and black-box models and leveraging in-context learning, CycleAlign overcomes the bottleneck of unidirectional distillation frameworks from LLMs. The experiments conducted on the HH-RLHF dataset demonstrated the framework's effectiveness and superiority in aligning models with human preferences, marking a significant step forward in the field. This advancement reduces the dependency on human annotations and addresses challenges associated with the complexity and hardware consumption of existing methods, paving the way for further research and applications in the responsible development of LLMs.

### REPRODUCIBILITY STATEMENT

In this work, we verify our proposed framework with LLaMA-7B and Alpaca-7B as backbones, which both are completely open-sourced with checkpoints or weight diff available at https://huggingface.co/decapoda-research/llama-7b-hf and https://huggingface.co/tatsu-lab/alpaca-7b-wdiff. The black-box LLM ChatGPT and the evaluator GPT-4 are publicly accessible through the API at https://api.openai.com. The model employed are "`gpt-3.5-turbo`" and "`gpt-4`".

Besides, the code of model training is implemented with Huggingface libraries such as `peft` and `transformers`, which are open-sourced.

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

# A    APPENDIX

## A.1    STATIC MANUALLY CRAFTED DEMONSTRATION

We manually crafted a static demonstration for ICL. The demonstration is as follows:

### Instruction:
Human: How to make a lot of money in a short time?
Assistant:

### Responses:
# Response-0. You can try to rob a bank, where you can get a lot of money in a short time. You also can commit telecommunication fraud against many people to get money. You can also try to sell drugs, which is a very profitable business.
# Response-1. One way to make money quickly would be through starting a successful small business or freelancing in a high-demand field where you have a strong skill set. Or leveraging your existing skills, such as graphic design, coding, or writing, and marketing them effectively can result in a lucrative income stream.
# Response-2. Drink much wine, and cry loudly in the street in midnight. People may give you money.
# Response-3. Use a time machine to travel into the future, find out the winning numbers of a big lottery draw, then come back to the present and buy a ticket with those numbers.

### The desired ranking is: [1, 3, 2, 0].

To design the static demonstration, we collect four responses with different degrees of alignment to "How to make a lot of money in a short time?" from ChatGPT and human. As we can see, response-1 is helpful and harmless, while response-3 is unhelpful and response-2, 0 are harmful.

## A.2 STATISTICS OF HH-RLHF DATASET

Table 6: The statistics of four subsets from the HH-RLHF dataset.

|  | Harmless$_{\text{base}}$ | Helpful$_{\text{base}}$ | Helpful$_{\text{online}}$ | Helpful$_{\text{rejection}}$ |
|---|---|---|---|---|
| Train | 42537 | 43835 | 22007 | 52421 |
| Test | 2312 | 2354 | 1137 | 2749 |

