# OpenReview forum: "CycleAlign: Iterative Distillation from Black-box LLM to White-box Models for Better Human Alignment"
_ICLR.cc/2024/Conference — ICLR 2024 Conference Withdrawn Submission_

### Official Review · Reviewer_NcP5 · 2023-10-28

**Soundness:** 1 poor
**Presentation:** 2 fair
**Contribution:** 2 fair
**Rating:** 3
**Confidence:** 4

**Summary:**

This paper proposes CycleAlign, an iteratively AI preference alignment framework. The core idea is collaborative interaction between the black-box LLM and white-box model. For each instruction, the white-box model generates multiple responses to an input prompt with varying degrees of human alignment. Then both white-box and black-box models rank the response at the same time, with the black-box one adopting some in-conext demonstration while white-box one doesn't. Then the white-box model is optimized based on the black-box ranking, while the demonstration of black-box model is updated with the ranking that both models agree. The cyclical interaction loops up to N times (N=5 in this paper). Experiments show the effectiveness of the dynamic alignment algorithm.

**Strengths:**

1. The dynamic alignment resembles the actor-critique framework of PPO to some extent. Therefore, compared to offline methods like DPO, RRHF, and PRO, the iterative update for the black-box model may keep it caught up with the knowledge/distribution of the white-box model and thus provide a more accurate ranking. This online idea is very interesting and the effectiveness of updating demonstration has been demonstrated in experiments.
2. This approach appears to be data efficient and works well with only 1000 instructions.

**Weaknesses:**

1. The biggest concern for the method is that, though the method is data-efficient, on the other hand, it can be expensive to scale up the instructions since each instruction requires querying commercial models 5 times. On the contrary, the baseline methods can easily scale up the training data and may benefit from larger data sizes.
2. The evaluation is skeptical. The author uses a reward model to score the responses of different models. However, the reliability and capacity of the reward model remain unknown, hence I am not sure if the evaluation is reliable. Moreover, only HH-RLHF is adopted for evaluation, which is not enough to demonstrate the general effectiveness.
3. The writing should be improved. For example, the author spent half of the abstraction to elicit the claim "researchers have begun to align the language model with human preference from AI feedback". On the contrary, their motivation is not well-explained. Also, as a very important metric in experiments, the details of the reward model they used is missing.
4. The method is not so effective regarding human evaluation. This method is comparable with the original PRO on helpfulness online/rejection split. This further confirms my concern about weakness #2.

**Questions:**

1. What is the "well-trained" reward model in evaluation? Can you explain it explicitly in the paper?
2. What dataset are you using in Fig 4 (sec 4.4)? Is it the same as Table 1 (sec 4.1)?

---

### Official Review · Reviewer_mkaA · 2023-10-30

**Soundness:** 3 good
**Presentation:** 3 good
**Contribution:** 3 good
**Rating:** 5
**Confidence:** 4

**Summary:**

The paper proposes CycleAlign that interactively improves black-box large model ranking results by incorporating white-box model ranking results in-contextly. To avoid the noises brought by white-box model ranking results, authors filter ranking results and use ranking results that achieve agreement between the two models. Moreover, multiple interactions per step are used to guarantee the performance. The paper investigates the effectiveness of CycleAlign on HH-RLHF dataset.

**Strengths:**

1. The interactive approach is novel and achieves good performance in experiments.
2. The paper is well-written and easy to understand.

**Weaknesses:**

My major concern is the computational overhead and the fair comparison. Compared with prior works, the proposed approach has multiple interactions (5 in this paper), which increases the training cost. My question is what will the vanilla approach work under the same computational cost (i.e., vanilla approach will annotate more data but will lower quality).

**Questions:**

1. Does the approach have a fixed number of interactions? I notice that the paper uses terminologies like "maximum interactive number" and "threshold of the interaction times." Does that mean authors use any techniques like early stopping? If so, what are the corresponding details?

2. The paper uses N to denote the interactive number under Section 2.1 and 4.4, and T in Section 3.3. Are they referring to the same hyper-parameter?

**Details Of Ethics Concerns:**

No ethics concerns are needed.

---

### Official Review · Reviewer_eDHy · 2023-11-01

**Soundness:** 2 fair
**Presentation:** 2 fair
**Contribution:** 2 fair
**Rating:** 3
**Confidence:** 4

**Summary:**

This paper proposes CycleAlign, a method to leverage a black-box LLM that well aligns with human preferences to improve the alignment of a white-box LLM. The black-box model performs in-context learning while the white-box model is finetuned for ranking-based alignment. the The black-box model predicts ranks that supervise the white model. while ranks that two models agree on are added to the in-context examples of the black-box model. Effectiveness is shown on a recent human preference dataset about helpfulness and harmlessness.

**Strengths:**

- The paper proposes a simple and intuitive method for using an aligned model to improve the alignment of another model.
- Clear positive results over the ranking-based alignment tuning baselines are shown on a recent benchmark.

**Weaknesses:**

- Not much method novelty. The main proposal is to add rankings that two models agree on to the in-context examples of the black-box model. However, agreement-based data filtering is common in the distillation literature. The paper does not provide sufficient justification about the uniqueness of the agreement-based data filtering in this case.
- The paper has not compared to recent RLAIF methods as mentioned by Reviewer q2L9.
- The writing should be proofread. There are typos like "Condiserind" and grammatical errors like "a well-trained model are".

**Questions:**

- Can there be negative mining methods such that we train the student using disagreed examples?
- Is teacher-student agreement necessary? If we let the teacher generate more samples and "self-rank", are those demonstrations similarly good?
- Can the finding that teacher-student agreement can be used to improve the teacher be generalized to more tasks or settings?

---

### Official Review · Reviewer_q2L9 · 2023-11-03

**Soundness:** 2 fair
**Presentation:** 2 fair
**Contribution:** 2 fair
**Rating:** 5
**Confidence:** 5

**Summary:**

This paper studied the alignment approach of large language models, focusing on reinforcement learning from AI feedback (RLAIF). The authors proposed a novel CycleAlign data augmentation method, which refines the raking of responses by utilizing both the teacher model (black box) and the student model (white box to be aligned) judgements. Then, the consistent ones are used as the demonstration for the black-box model and dynamically updated during the training process. The effectiveness of CycleAlign is tested on two popular open-source LLMs, LLaMA and Alpaca, showing further improvements upon the original SFT alignment methods.

**Strengths:**

* This paper took a step towards addressing a critical problem, the high data/training cost of LLM alignment and proposed a lightweight RLAIF method, which is more suitable for real scenarios.

* The idea of combining the rankings from both teacher and student models is novel. The experimental results also demonstrate significant improvements upon the original SFT-based alignment methods.

**Weaknesses:**

Though this paper is relatively well-written and organized, there are three main problems:

* The motivation of the core components is questionable, and some key information is missing. Since RLAIF is not a new idea, the main contributions of this work lie in (a) the combination of both student and teacher models’ rankings and (b) dynamical demonstrations.
    1. The authors claimed several times that their method could ‘*break the bottleneck of black-box models*’ (Sec. 1, Sec. 4.3, Sec. 7), but it’s unclear what the bottleneck is and why the proposed method would work for it.

    2.  (a) is unreasonable and unnatural for me. Since the black-box model is always weaker than the black-box one (ChatGPT in this work) (as shown in Table 1), the white-box LLM’s ranking would bring more noise. It’s hard to understand why it could improve the alignment process.

* The experimental settings are problematical, making results unconvincing.

    1. The proposed method, as a kind of RLAIF, should be compared with previous RLAIF methods (Bai et al., 2022; Wang et al., 2023; Sun et al., 2023; Lee et al., 2023; Yang et al., 2023), instead of the original ranking-based alignment tuning methods. Since RLAIF can be considered a kind of data augmentation method, it would definitely improve the original methods.

    2. Some key ablation experiments are missing. To justify the effectiveness of the two designs (a) and (b), besides the ablation on dynamic demonstration, the authors should also compare more settings: i) with only the black-box rankings and ii) with only white-box rankings.

* Some essential designs need more in-depth analysis.
   1. The performance drop of CycleAlign_{PRO} + Alpaca and LLaMa on Harmless_{base} in Table 1.
   2. In Table 4, CycleAlign+PRO w/o dynamic demonstration achieves a total reward of 67.36, even worse than the original PRO (67.64) in Table 1. What is the reason?

References:
* Bai et al., Constitutional AI: Harmlessness from AI Feedback. 2022.
* Wang et al., SELF-INSTRUCT: Aligning Language Models with Self-Generated Instructions. 2023.
* Lee et al., RLAIF: Scaling Reinforcement Learning from Human Feedback with AI Feedback. 2023.
* Yang et al., Reinforcement Learning from Contrast Distillation for Language Model Alignment. 2023.

Missing Reference:
* Sun et al., Principle-Driven Self-Alignment of Language Models from Scratch with Minimal Human Supervision. 2023.

**Questions:**

1. What is ‘the bottleneck in the ranking performance of the black-box models’?
2. How to explain the performance drop of CycleAlign_{PRO} + Alpaca and LLaMa on Harmless_{base} in Table 1?